# Reduction strategies for inpatient oral third-generation cephalosporins at a cancer center: An interrupted time-series analysis

**Naoya Itoh**[1,2]*, **Takanori Kawabata**[3], **Nana Akazawa**[2], **Daichi Kawamura**[2], **Hiromi Murakami**[2], **Yuichi Ishibana**[2], **Eiichi N. Kodama**[4], **Norio Ohmagari**[1,5,6]

1 Collaborative Chairs Emerging and Reemerging Infectious Diseases, National Center for Global Health and Medicine, Graduate School of Medicine, Tohoku University, Miyagi, Japan, 2 Division of Infectious Diseases, Aichi Cancer Center Hospital, Nagoya, Japan, 3 Department of Data Science, National Cerebral and Cardiovascular Center, Suita, Japan, 4 Division of Infectious Diseases, International Research Institute of Disaster Science, and Graduate School of Medicine, Tohoku University and Tohoku Medical Megabank Organization, Sendai, Japan, 5 AMR Clinical Reference Center, Disease Control and Prevention Center, National Center for Global Health and Medicine, Tokyo, Japan, 6 Disease Control and Prevention Center, National Center for Global Health and Medicine, Tokyo, Japan

* itohnaoya0925@ybb.ne.jp

**Data Availability Statement:** All relevant data are within the manuscript and its Supporting Information files.

## Abstract

Oral third-generation cephalosporins (3GCs) are not recommended for use owing to their low bioavailability and the risk of emergence of resistant microorganisms with overuse. A standardized and effective method for reducing their use is lacking. Here, in a 60-month, single-institution, interrupted time-series analysis, which was retrospectively conducted between April 1, 2017, and March 31, 2022, we evaluated the effectiveness of a four-phase intervention to reduce the use of 3GCs in patients at a cancer center: Phase 1 (pre-intervention), Phase 2 (review of clinical pathways), Phase 3 (establishment of infectious disease consultation service and implementation of antimicrobial stewardship program), and Phase 4 (educational lecture and pop-up displays for oral antimicrobials at the time of ordering). Although no significant changes were observed in Phases 3 and 4, the first intervention resulted in a significant decrease in the trend and level of days of therapy (DOT) for 3GCs. The level for cephalexin DOT and the trend for sulfamethoxazole-trimethoprim DOT increased in Phase 4, and the trend for amoxicillin and amoxicillin-clavulanate DOT increased in Phase 3. Macrolide DOT showed a decreasing trend in Phases 2 and 4 and decreasing and increased levels in Phases 3 and 4, respectively; no change was observed for quinolones. Actual and adjusted purchase costs of 3GCs decreased significantly during all study periods, while those for oral antimicrobials decreased in Phase 2, and actual purchase costs increased in Phases 3 and 4. No significant reduction in resistant organisms, length of hospital stay, or mortality was observed. This is the first study on the effects of oral 3GC reduction strategies in patients with cancer. We conclude that even facilities that substantially use antimicrobials can efficiently reduce the use of 3GCs.

**Funding:** This work was supported by grants from the National Academic Research Grant Funds [JSPS KAKENHI: 22K10547] and the Aichi Cancer Research Foundation. The funder had no role in the study design, data collection, analysis, or manuscript preparation.

**Competing interests:** The authors have declared that no competing interests exist.

## Introduction

The spread of drug-resistant bacteria is a serious global health threat [1]. In Japan, as in other countries, there is concern about the relationship between inappropriate use of antimicrobial agents and antimicrobial resistance [2]. Patients with cancer often receive multiple antimicrobial treatments following chemotherapy or surgery owing to the progression of the underlying disease or complications, resulting in substantial antimicrobial use [3, 4]. In patients with cancer, appropriate diagnosis and management of infections are often challenging because patients often present with subtle or atypical symptoms, and inappropriate antimicrobial use is frequent [3–5]. Therefore, there is an urgent need to address the increasing number of antimicrobial-resistant bacteria and prevent their occurrence in patients with cancer.

A survey conducted from 2009 to 2013 in Japan reported that oral antimicrobial consumption accounted for 92.6% of the total antimicrobial consumption [6]. The consumption of oral third-generation cephalosporins (3GCs), macrolides, and fluoroquinolones was specifically higher in Japan than in the United States and European countries [6–8]. Japan announced the National Action Plan (NAP) for antimicrobial resistance in April 2016, with the specific goal of reducing the use of antimicrobials by 2020 [2]. In addition to total antimicrobial use, one of the targets established in the NAP was a 50% reduction in the use of oral 3GCs by 2020; however, the reduction rate of 3GCs in Japan at the time was only 36.39% [9].

Oral 3GCs are not widely recommended owing to several reasons. First, the selective pressure from the extensive use of oral antimicrobials in Japan, including 3GCs, may have led to the emergence of resistant bacteria [6]. Inappropriate use of oral 3GCs has been implicated as a cause for the higher proportion of β-lactamase-nonproducing ampicillin-resistant *Haemophilus influenzae* (BLNAR) and penicillin-resistant *Streptococcus pneumoniae* (PRSP) in Japan compared with that in European countries and the United States [2, 10–13]. In addition, extended-spectrum β-lactamase (ESBL)-producing Enterobacteriaceae counts are increasing in Japan [14, 15], and Hosokawa et al. reported that the use of oral 3GCs is responsible for the increase in ESBL-producing *Escherichia coli* [16]. The oral bioavailability of the oral 3GCs cefditoren and cefdinir is low (14% and 25%, respectively) [17], which may lead to an increase in drug-resistant strains [16]. Cephalosporins are also known to be one of the antimicrobial agents most likely to cause *Clostridioides difficile* infection [18]. Furthermore, 3GCs such as cefditoren and cefcapene possess pivoxil groups and have been reported to cause hypocarnitinemia in patients receiving these drugs [19, 20].

Although efforts to reduce the use of oral 3GCs in Japan have been reported, a uniform method of reduction has not been established [21–24]. Notably, there are no studies on antimicrobial stewardship programs (ASPs) reported on oral 3GCs in countries outside Japan. Furthermore, there are no studies aimed at reducing the use of oral 3GCs in patients with cancer.

Here, we conducted a clinical path review including oral 3GCs, implemented an infectious disease (ID) consultation and ASP, conducted an educational lecture on oral antimicrobials, and introduced an alert system when prescribing oral 3GCs at a cancer center in Japan. The effects of these interventions on the use of oral 3GCs and other oral antimicrobials in hospitalized patients, the prevalence of resistant organisms, the cost of oral antimicrobials, and the patient outcomes were evaluated.

## Materials and methods

### Setting

This study was conducted at the 500-bed tertiary-care Aichi Cancer Center (ACC) Hospital in Aichi, Japan. This hospital contains 23 clinical departments and admits approximately 11,000

patients per year. There are 15 departments in charge of inpatients, including the following: Plastic and Reconstructive Surgery, Hematology and Cell Therapy, Thoracic Surgery, Thoracic Oncology, Gastroenterological Surgery, Gastroenterology, Orthopedic Surgery, Head and Neck Surgery, Breast Oncology, Neurosurgery, Urology, Gynecologic Oncology, Radiation Oncology, Diagnostic and Interventional Radiology, and Clinical Oncology.

## Study design

This study was a single-institution retrospective, observational, interrupted time-series analysis conducted during a 60-month period between April 1, 2017, and March 31, 2022. All data for this study were obtained from the ACC Hospital database; microorganism data were procured from the microbiology laboratory, and prescription data were procured from the pharmacy department.

## Interventions for reducing the use of oral 3GCs

The intervention evaluated in this study was implemented in four phases:

Phase 1: Pre-intervention period (April 1, 2017, to May 31, 2019)

Phase 2: Review of clinical pathways (June 1, 2019, to March 31, 2020)
Beginning in June 2019, cefdinir was removed from the orthopedic perioperative clinical pathway, and cefditoren was removed from the hospital's list of adopted drugs.

Phase 3: Establishing an ID consultation service and implementing ASP (April 1, 2020, to June 30, 2021)
The ID consultation and ASP were introduced on April 1, 2020. The ID consultation service is a system in which, for 5 days per week, full-time ID physicians provide ID consultation for referrals from the other 15 departments, review positive blood culture results to ensure that patients receive the appropriate empirical treatment, and provide feedback to other physicians. The ASP was structured as follows: for 3 days per week, the antimicrobial stewardship team evaluated medical records and antimicrobial use for patients who were administered specific intravenous broad-spectrum antibiotics (vancomycin, teicoplanin, daptomycin, linezolid, cefepime, cefozopran, piperacillin–tazobactam, imipenem–cilastatin, meropenem, and doripenem) and provided post-prescription review with feedback to the physician (changed to 5 days per week from April 2021 and addition of intravenous levofloxacin). In October 2020, cefdinir was removed from the list of drugs adopted in hospitals.

Phase 4: Educational lecture and pop-up displays for oral antimicrobials (July 1, 2021, to March 31, 2022)
Lectures on oral antimicrobial agents were conducted for all healthcare professionals, including physicians, nurses, and pharmacists, between July 1, 2021 and July 31, 2021, with a 100% attendance rate. From November 1, 2021, a pop-up was displayed on the electronic medical record when prescribing third-generation oral antimicrobials (**Table 1**).

## Primary outcome measures

The primary outcome measured was the change in days of therapy (DOT) with oral 3GCs (3GCs-DOT; for cefdinir, cefcapene, and cefditoren), expressed as DOT per 100 patient-days per month.

**Table 1. Pop-up for oral 3GC prescription.**

| |
|---|
| Oral third-generation cephalosporins are not recommended due to their low bioavailability, as well as problems with drug-resistant bacteria. Please consider switching to another drug. |
| Examples of switching to other drugs: <br> 1) Oral switch from cefazolin: cefalexin <br> 2) Oral switch from ampicillin-sulbactam or cefmetazole: amoxicillin-clavulanate <br> 3) Empirical treatment for cystitis: cefalexin or sulfamethoxazole-trimethoprim |
| If you have any questions about oral antimicrobial prescription, please contact the Department of Infectious Diseases. |

## Secondary outcome measures

**DOT for cefalexin, amoxicillin, amoxicillin-clavulanate, and sulfamethoxazole-trimethoprim.** The total DOT per month per 100 patient-days was calculated for the four antimicrobial agents cefalexin (first-generation cephalosporin), amoxicillin, amoxicillin-clavulanate, and sulfamethoxazole-trimethoprim, as these antibiotics are narrow-spectrum antimicrobials. This parameter was assessed to confirm whether the intervention was performed appropriately.

**DOT for quinolones and macrolides.** The total DOT per month per 100 patient-days was calculated for three quinolones (moxifloxacin, ciprofloxacin, and levofloxacin) and three macrolides (erythromycin, azithromycin, and clarithromycin). This was because these antibiotics are broad-spectrum antimicrobials, similar to 3GCs. The DOT for quinolones and macrolides was evaluated to assess whether 3GCs were simply being replaced with other broad-spectrum agents.

## Incidence of hospital-acquired infection with resistant microorganisms and *C. difficile*

We measured the annual incidence of infections with antibiotic-resistant bacteria and *C. difficile* infection (CDI) per 1000 patient-days between April 2017 and March 2022, the period for which these data were available, as an indicator of the outcome of our interventions. The resistant microorganisms included ESBL-producing Enterobacteriaceae, PRSP, and resistant *Haemophilus influenzae*. Hospital-acquired microorganisms were defined as those that were identified more than 72 h after admission [25]. To exclude duplication, when the same resistant microorganisms were isolated more than once in the same patient, only the first specimen obtained each month was included in the analysis [25]. However, if the resistant microorganism was detected in a blood specimen, this infection was defined as a new episode if the same resistant bacterium had not been detected in blood samples from the same patient within the previous two weeks [25]. Only clinical specimens of resistant microorganisms were included, and specimens for surveillance culture and negative confirmation were excluded. CDI was defined as the number of patients with positive CD toxin results (C. DIFF QUIK CHEK COMPLETE; Alere Medical Co., Tokyo, Japan). PRSP was defined as PCG minimum inhibitory concentration (MIC) of $\geq 0.12$ μg/mL according to the CLSI 2012 criteria (parenteral penicillin, meningitis) [26]. Resistant *H. influenzae* was defined as follows: MIC of ABPC $\geq 4.0$ μg/mL and MIC of ampicillin-sulbactam $\geq 4/2$ μg/mL or amoxicillin-clavulanic acid $\geq 8/4$ according to the CLSI 2021 criteria [27]. ESBL-producing Enterobacteriaceae, including *E. coli*, *Klebsiella pneumoniae*, *Klebsiella oxytoca*, and *Proteus mirabilis*, were identified using the Cica-β test (Kanto Chemical Co., Tokyo, Japan) and disc diffusion method.

### Cost of 3GCs and all oral antimicrobials

To assess the economic impact of our interventions, we assessed the cost of purchasing antimicrobials per patient-day each year from April 2017 to March 2022 and April 2018 to March 2021. The exchange rate of 1 USD to 108 JPY was used for the calculations in March 2021.

### Total number of inpatient specimens

To assess the influence of the total number of specimens on the detection of resistant organisms, we evaluated the total number of inpatient specimens per 100 patient-days from April 2017 to March 2022.

### All-cause in-hospital mortality and length of hospital stay

Data for all-cause in-hospital mortality and length of hospital stay were extracted and included in the analysis.

### Statistical analysis

To demonstrate the effect of each intervention on 3GCs-DOT, the DOT for cefalexin, amoxicillin, amoxicillin-clavulanate, sulfamethoxazole-trimethoprim, the three quinolones (moxifloxacin, ciprofloxacin, and levofloxacin), and the three macrolides (erythromycin, azithromycin, and clarithromycin) was calculated. Furthermore, we carried out a segmented regression analysis of interrupted time-series studies for the four periods (Phase 1: April 1, 2017, to May 31, 2019; Phase 2: June 1, 2019, to March 31, 2020; Phase 3: April 1, 2020, to June 30, 2021; Phase 4: July 1, 2021, to March 31, 2022), when DOT data were available. Trends and changes in the levels of incidence of antibiotic-resistant bacterial infection, CDI, all-cause in-hospital mortality, and length of hospital stay were evaluated using a segmented regression analysis of interrupted time-series studies for two periods (Phase 1, Phases 2–4). We adopted a linear regression model with the Prais–Winsten estimator using Generalized Least Squares. Seasonality was not statistically confirmed to affect the primary outcome. This model was also used for all secondary outcomes. Bivariate analysis for the cost of antimicrobials was carried out using the Mann–Whitney U test (continuous variables), with $p < 0.05$ regarded as statistically significant. The R software, version 4.0.2. (The R Foundation for Statistical Computing, Vienna, Austria) was used for all analyses.

### Ethical considerations

This study was approved by the Institutional Review Board of the ACC Hospital (approval number: 2021-0-176) and conducted according to the principles of the Declaration of Helsinki. The requirement of informed consent was waived because this study only used data that were collected in clinical practice.

## Results

### Use of 3GCs

Changes in 3GCs-DOT during the four phases are shown in **Fig 1**. After initiation of the first intervention, the monthly 3GCs-DOT showed a decreasing trend (coefficients: -0.08; 95% confidence interval [CI]: -0.15 to -0.01, $p < 0.05$) and the level of the monthly 3GCs-DOT also reduced (coefficients: -0.5; 95% CI: -0.93 to 0.03, $p < 0.05$). There were no significant changes in trends and levels from Phases 2 to 3 or from Phases 3 to 4.

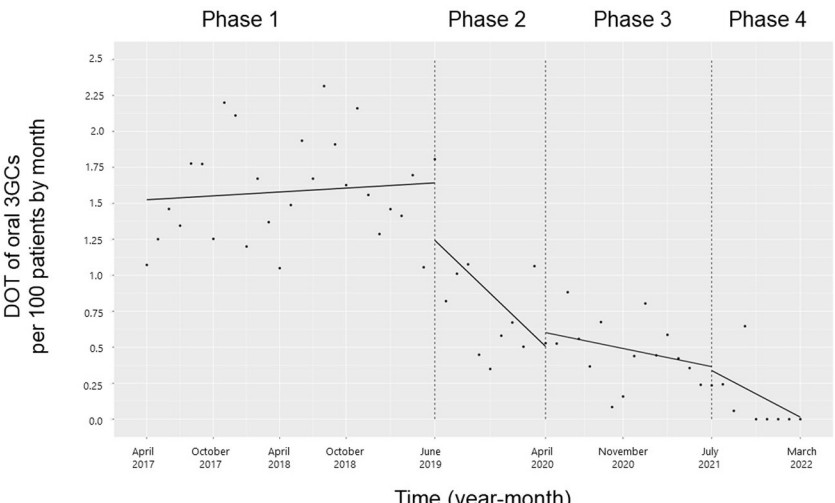

**Fig 1. Trends in the days of third-generation cephalosporin therapy per 100 patients, by month, during Phases 1 to 4.** Each dot refers to the third-generation cephalosporins (3GCs) per 100 patients each month, and the slope is based on the linear regression in the four phases: Phase 1 (pre-intervention period from April 1, 2017, to May 31, 2019); Phase 2 (review of clinical pathways from June 1, 2019, to March 31, 2020); Phase 3 (establishing an infectious disease (ID) consultation service and implementing the antimicrobial stewardship program (ASP) from April 1, 2020, to June 30, 2021); Phase 4 (educational lecture and pop-up displays for oral antimicrobials from July 1, 2021, to March 31, 2022).

## Use of cefalexin, amoxicillin, amoxicillin-clavulanate, and sulfamethoxazole-trimethoprim

There were no significant changes in trends and levels for cephalexin from Phases 1 to 2 or from Phases 2 to 3. The trend in the monthly DOT of cephalexin did not change from Phases 3 to 4, but the level increased (coefficients: 0.42; 95% CI: 0.03 to 0.81, p = 0.04; **Fig 2**).

There were no significant changes in the trend and level of the monthly DOT of amoxicillin and amoxicillin-clavulanate from Phases 1 to 2 (**Fig 3**). The trend in the monthly DOT of amoxicillin and amoxicillin-clavulanate increased from Phases 2 to 3 (coefficients: 0.26; 95% CI: 0.07 to 0.45, p = 0.01) and decreased from Phases 3 to 4 (coefficients: -0.67; 95% CI: -0.89 to -0.45, p < 0.001), while the level did not change.

The trend and level of the monthly DOT of sulfamethoxazole-trimethoprim did not change significantly from Phases 1 to 2 or Phases 2 to 3 (**Fig 4**). However, the trend in the monthly DOT of sulfamethoxazole-trimethoprim increased from Phases 3 to 4 (coefficients: 0.26; coefficient: 0.38; 95% CI: 0.07 to 0.70, p = 0.02), while the level did not change.

## Use of quinolone and macrolide

The trend and level of the monthly DOT of all three quinolones (moxifloxacin, ciprofloxacin, and levofloxacin) did not change significantly during the study period (**Fig 5**).

The trend in the monthly DOT of the three macrolides (erythromycin, azithromycin, and clarithromycin) decreased from Phases 1 to 2 (coefficients: -0.13; 95% CI: -0.22 to -0.04, p < 0.01), but the level did not change (**Fig 6**). The trend in the monthly DOT of macrolides did not change from Phases 2 to 3 or from Phases 3 to 4, but the level decreased (coefficient: -0.96; 95% CI: -1.62 to -0.30, p < 0.01) and increased (coefficients: 0.92; 95% CI: 0.28 to 1.57, p < 0.01), respectively.

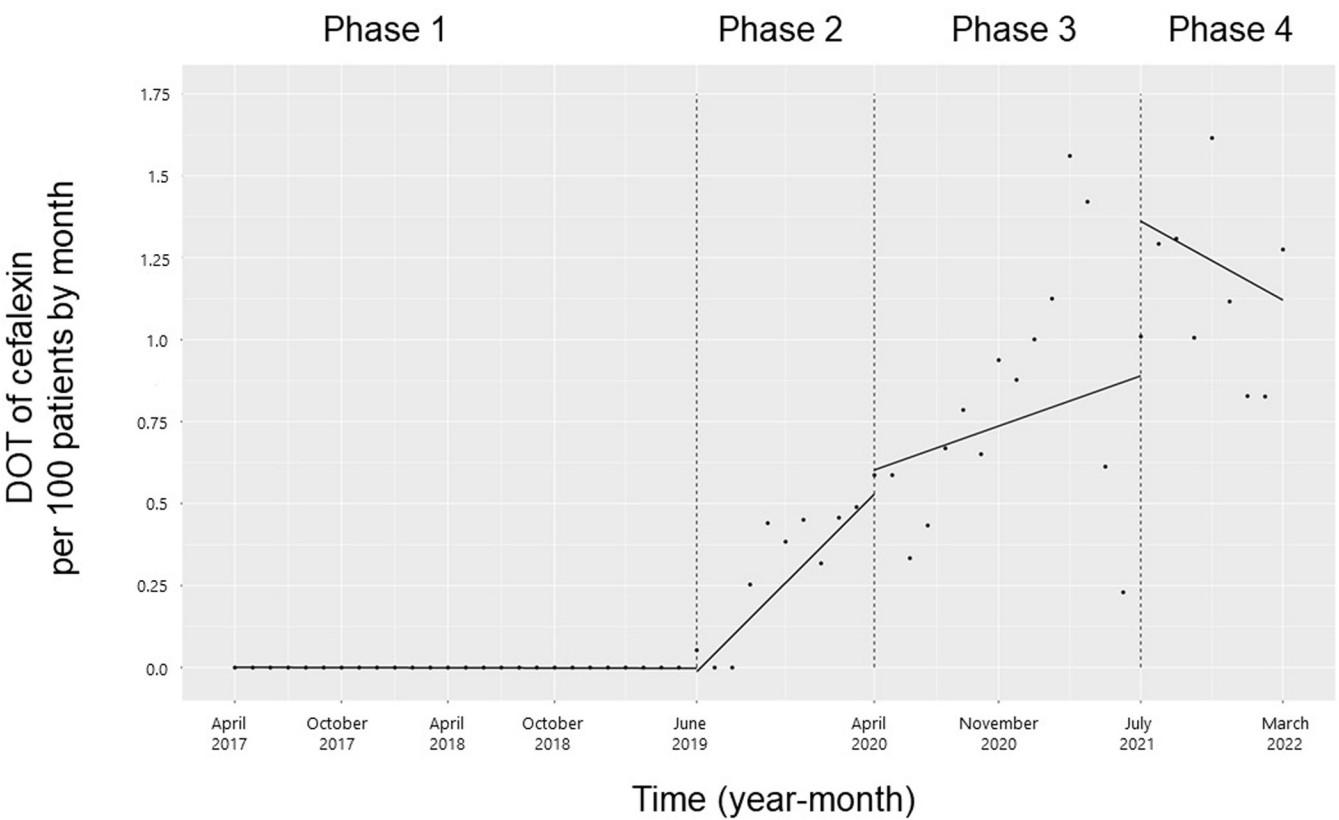

**Fig 2. Trends in the days of cefalexin therapy per 100 patients, by month, during Phases 1 to 4.** Each dot refers to cefalexin per 100 patients in each month, and the slope is based on the linear regression in the four phases: Phase 1 (pre-intervention period from April 1, 2017, to May 31, 2019); Phase 2 (review of clinical pathways from June 1, 2019, to March 31, 2020); Phase 3 (establishing an ID consultation service and implementing the ASP from April 1, 2020, to June 30, 2021); and Phase 4 (educational lecture and pop-up displays for oral antimicrobials from July 1, 2021, to March 31, 2022).

## Incidence of infection with antimicrobial-resistant microorganisms and CDI

The trend and level of the monthly incidence of ESBL-producing Enterobacteriaceae, PRSP, resistant *H. influenzae* and CDI did not change significantly during the study period (**Figs 7–10**). The trend in the monthly incidence of methicillin-resistant *Staphylococcus aureus* (MRSA) increased during the study period (coefficients: 0.05; 95% CI: 0.04 to 0.07, $p < 0.001$), but the level did not change (**Fig 11**).

## Cost of 3GCs and all oral antimicrobials

**Table 2** shows the actual and adjusted average costs of 3GCs and all oral antimicrobials. The actual (Phase 1 to 2: $p < 0.001$; Phase 2 to 3: $p < 0.01$, Phase 3 to 4: $p < 0.001$) and adjusted 3GC purchase costs (Phase 1 to 2: $p < 0.001$; Phase 2 to 3: $p < 0.01$, Phase 3 to 4: $p < 0.01$) per patient-days significantly decreased during the study period. The actual purchasing cost of all oral antimicrobials per patient-day decreased significantly from Phases 1 to 2 ($p < 0.01$) but increased from Phases 2 to 3 ($p < 0.01$) and from Phases 3 to 4 ($p = 0.01$). The adjusted purchase cost of all oral antimicrobials per patient-day decreased significantly from Phases 1 to 2 ($p < 0.001$) but did not change from Phases 2 to 3 ($p = 0.18$) or from Phases 3 to 4 ($p = 0.29$).

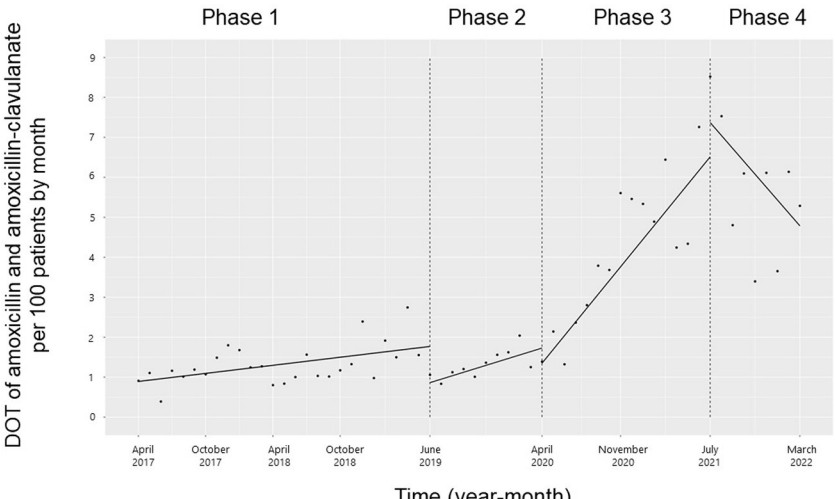

**Fig 3. Trends in the days of amoxicillin and amoxicillin-clavulanate therapy per 100 patients, by month, during Phases 1 to 4.** Each dot refers to the amoxicillin and amoxicillin-clavulanate per 100 patients each month, and the slope is based on the linear regression in the four phases: Phase 1 (pre-intervention period from April 1, 2017, to May 31, 2019); Phase 2 (review of clinical pathways from June 1, 2019, to March 31, 2020); Phase 3 (establishing an ID consultation service and implementing the ASP from April 1, 2020, to June 30, 2021); and Phase 4 (educational lecture and pop-up displays for oral antimicrobials from July 1, 2021, to March 31, 2022).

## Total number of inpatient specimens

The number of inpatient samples per 100 patient-day did not change from Phases 1 to 2 (median: 5.05 vs. 4.85, p = 0.412) but increased from Phases 2 to 3 (median: 4.85 vs. 6.46, p < 0.001). There was no change in the number of inpatient samples in Phases 3 to 4 (median: 6.46 vs. 6.95, p = 0.07).

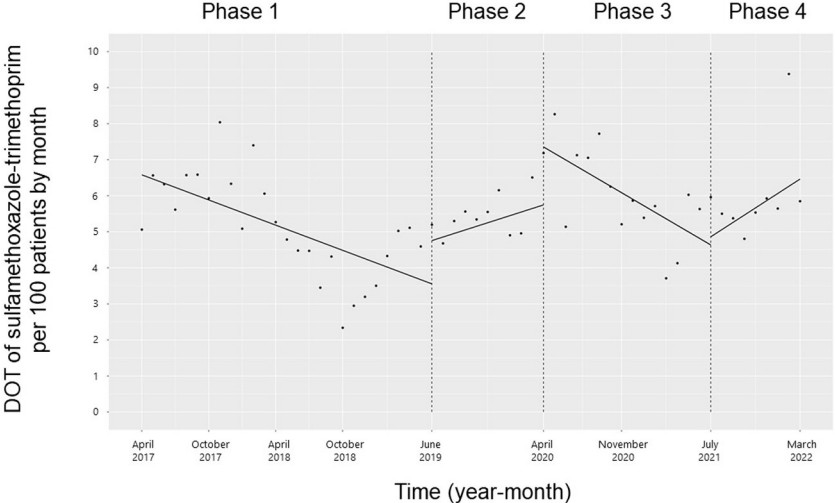

**Fig 4. Trends in the days of sulfamethoxazole-trimethoprim therapy per 100 patients, by month, during Phases 1 to 4.** Each dot refers to the sulfamethoxazole-trimethoprim per 100 patients each month, and the slope is based on the linear regression in the four phases: Phase 1 (pre-intervention period from April 1, 2017, to May 31, 2019); Phase 2 (review of clinical pathways from June 1, 2019, to March 31, 2020); Phase 3 (establishing an ID consultation service and implementing the ASP from April 1, 2020, to June 30, 2021); and Phase 4 (educational lecture and pop-up displays for oral antimicrobials from July 1, 2021, to March 31, 2022).

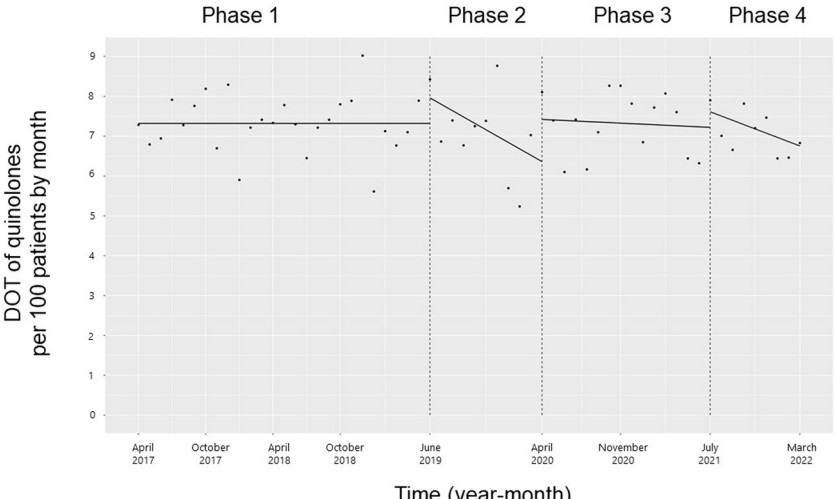

**Fig 5. Trends in the days of quinolone therapy per 100 patients, by month, during Phases 1 to 4.** Each dot refers to the quinolones per 100 patients in each month, and the slope is based on the linear regression in the four phases: Phase 1 (pre-intervention period from April 1, 2017, to May 31, 2019); Phase 2 (review of clinical pathways from June 1, 2019, to March 31, 2020); Phase 3 (establishing an ID consultation service and implementing the ASP from April 1, 2020, to June 30, 2021); and Phase 4 (educational lecture and pop-up displays for oral antimicrobials from July 1, 2021, to March 31, 2022).

## All-cause in-hospital mortality and length of hospital stay

There was no significant change in the trend of in-hospital mortality, length of hospital stay, or their level (**Figs 12** and **13**).

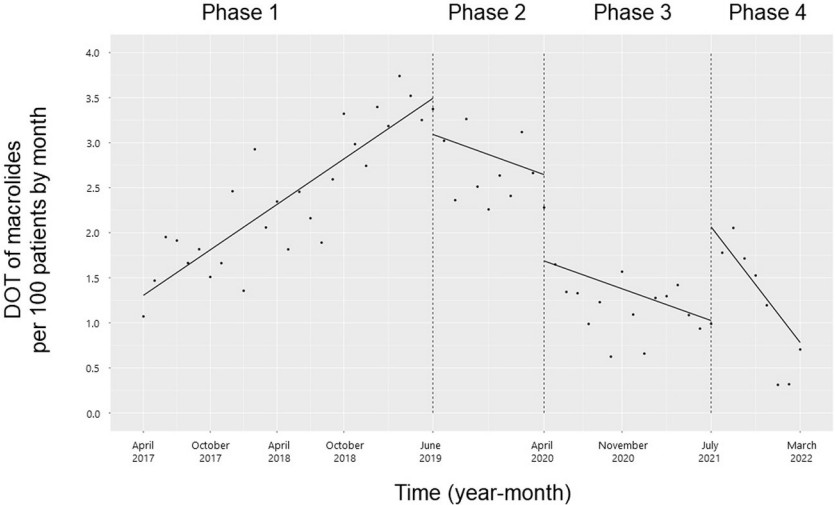

**Fig 6. Trends in the days of macrolides therapy per 100 patients, by month, during Phases 1 to 4.** Each dot refers to the macrolides per 100 patients each month, and the slope is based on the linear regression in the four phases: Phase 1 (pre-intervention period from April 1, 2017, to May 31, 2019); Phase 2 (review of clinical pathways from June 1, 2019, to March 31, 2020); Phase 3 (establishing an ID consultation service and implementing the ASP from April 1, 2020, to June 30, 2021); and Phase 4 (educational lecture and pop-up displays for oral antimicrobials from July 1, 2021, to March 31, 2022).

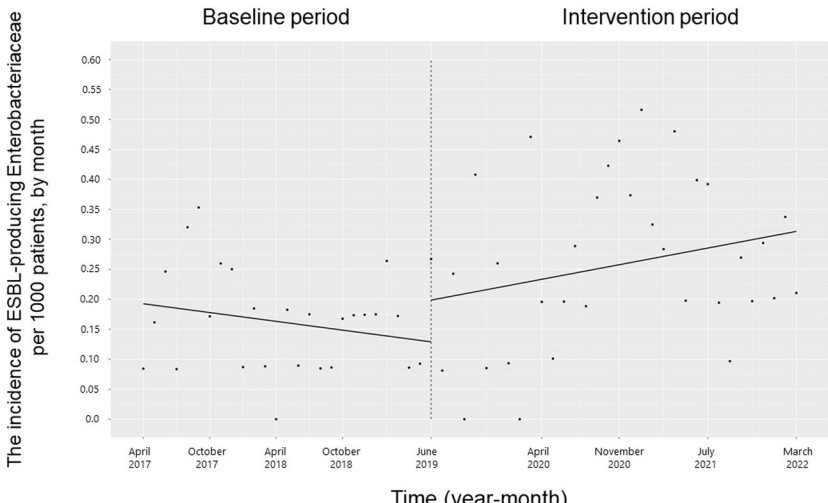

**Fig 7. Trends in the incidence of extended-spectrum β-lactamase (ESBL)- producing Enterobacteriaceae per 1000 patients, by month, during the pre-intervention period (Phase 1) and interventional period (Phases 2–4).** Each dot refers to ESBL-producing Enterobacteriaceae per 1000 patients in each month; the slope is based on linear regression in the four phases: Phase 1 (pre-intervention period from April 1, 2017, to May 31, 2019); Phase 2 (review of clinical pathways from June 1, 2019, to March 31, 2020); Phase 3 (establishing an infectious disease (ID) consultation service and implementing the antimicrobial stewardship program from April 1, 2020, to June 30, 2021); and Phase 4 (educational lecture and pop-up displays for oral antimicrobials from July 1, 2021, to March 31, 2022).

## Discussion

The present study is the first to report the effects of an oral 3GC reduction strategy in patients with cancer. It is difficult to provide comprehensive interventions in cancer centers owing to

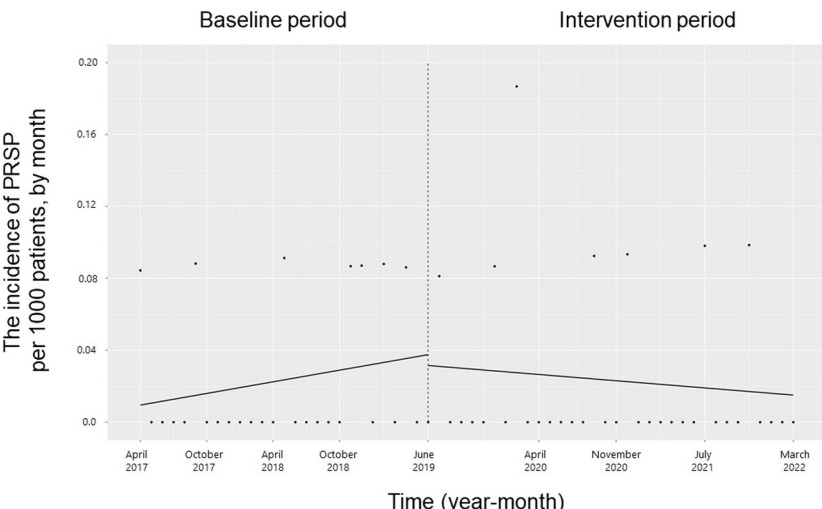

**Fig 8. Trends in the incidence of penicillin-resistant *Streptococcus pneumoniae* (PRSP) per 1000 patients, by month, during the pre-intervention period (Phase 1) and interventional period (Phases 2–4).** Each dot refers to the PRSP per 1000 patients in each month; the slope is based on linear regression in the four phases: Phase 1 (pre-intervention period from April 1, 2017, to May 31, 2019); Phase 2 (review of clinical pathways from June 1, 2019, to March 31, 2020); Phase 3 (establishing an infectious disease (ID) consultation service and implementing the antimicrobial stewardship program from April 1, 2020, to June 30, 2021); and Phase 4 (educational lecture and pop-up displays for oral antimicrobials from July 1, 2021, to March 31, 2022).

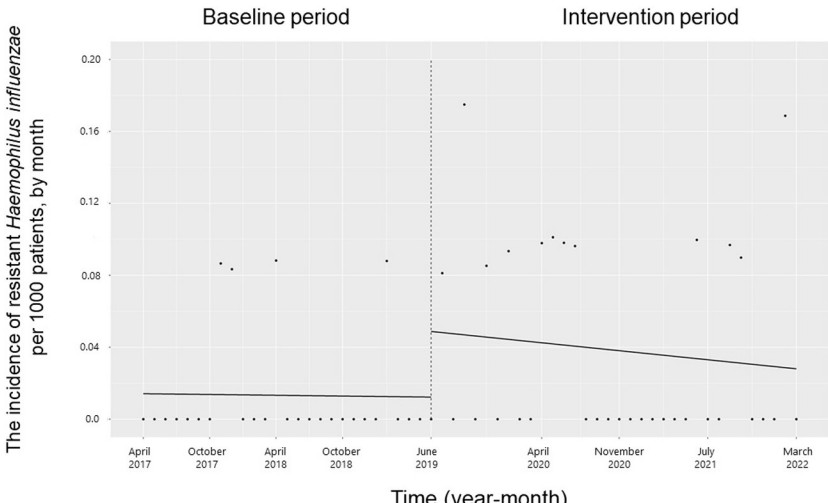

**Fig 9. Trends in the incidence of resistant *Haemophilus influenzae* per 1000 patients, by month, during the pre-intervention period (Phase 1) and interventional period (Phases 2–4).** Each dot refers to resistant *H. influenzae* per 1000 patients each month; the slope is based on linear regression in the four phases: Phase 1 (pre-intervention period from April 1, 2017, to May 31, 2019); Phase 2 (review of clinical pathways from June 1, 2019, to March 31, 2020); Phase 3 (establishing an infectious disease consultation service and implementing the antimicrobial stewardship program from April 1, 2020, to June 30, 2021); and Phase 4 (educational lecture and pop-up displays for oral antimicrobials from July 1, 2021, to March 31, 2022).

the complexity of patient conditions caused by underlying cancers, hematological malignancies, neutropenia, various departments involved in multidisciplinary patient management, departmental structures, and internal guidelines [28]. In addition, the use of antimicrobials is higher among patients with cancer than among the general population [3, 4]. However, even

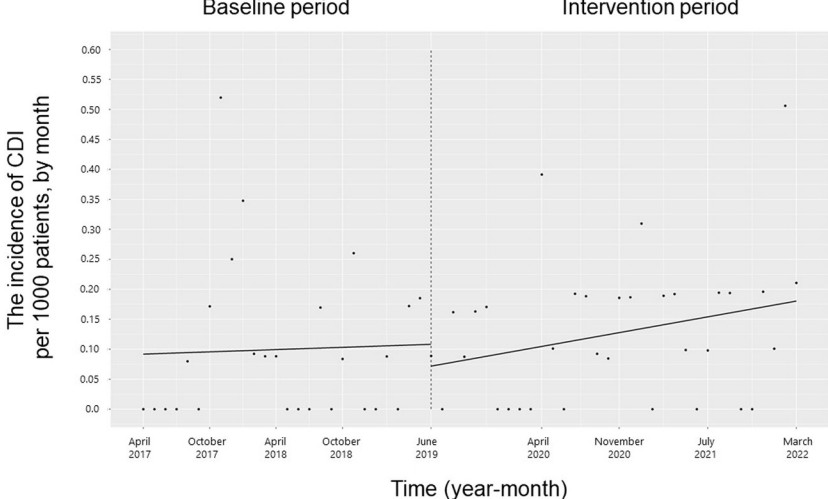

**Fig 10. Trends in the incidence of *Clostridioides difficile* infection (CDI) per 1000 patients, by month, during the pre-intervention period (Phase 1) and interventional period (Phases 2–4).** Each dot refers to the CDI per 1000 patients each month, and the slope is based on linear regression in the four phases: Phase 1 (pre-intervention period from April 1, 2017, to May 31, 2019); Phase 2 (review of clinical pathways from June 1, 2019, to March 31, 2020); Phase 3 (establishing an infectious disease (ID) consultation service and implementing the antimicrobial stewardship program from April 1, 2020, to June 30, 2021); and Phase 4 (educational lecture and pop-up displays for oral antimicrobials from July 1, 2021, to March 31, 2022).

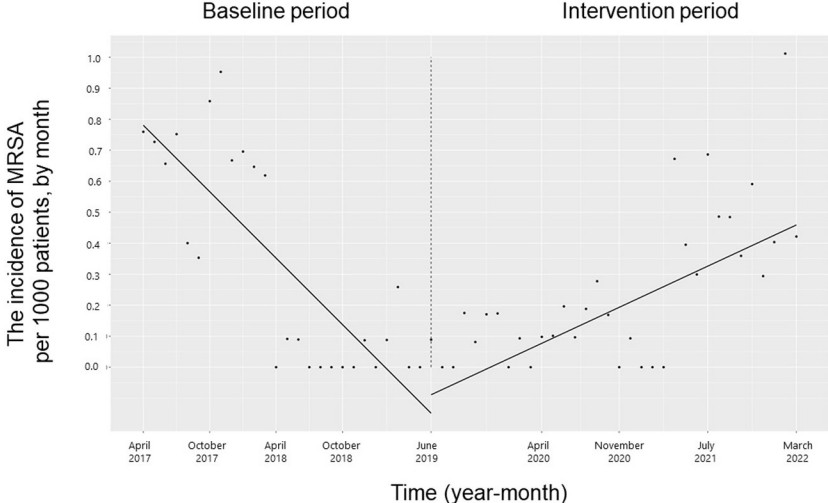

**Fig 11. Trends in the incidence of methicillin-resistant *Staphylococcus aureus* (MRSA) per 1000 patients, by month, during the pre-intervention period (Phase 1) and interventional period (Phases 2–4).** Each dot refers to MRSA per 1000 patients each month, and the slope is based on linear regression in the four phases: Phase 1 (pre-intervention period from April 1, 2017, to May 31, 2019); Phase 2 (review of clinical pathways from June 1, 2019, to March 31, 2020); Phase 3 (establishing an infectious disease (ID) consultation service and implementing the antimicrobial stewardship program from April 1, 2020, to June 30, 2021); and Phase 4 (educational lecture and pop-up displays for oral antimicrobials from July 1, 2021, to March 31, 2022).

in a population of patients with cancer and high antimicrobial use, each of our interventions contributed to a reduction in the use of oral 3GCs without worsening patient outcomes or increasing the use of alternative broad-spectrum antimicrobial agents and decreased the overall cost of oral antimicrobial agents.

In the current study, 3GCs showed a significant decrease in trends and levels with the clinical path review (Phase 2). Subsequently, both trends and levels showed a decline, but not significantly. This finding suggested that Phases 3 and 4 maintained the intervention effects from Phase 2 and implied that the mandatory intervention of clinical path review had a significant impact on the overall 3GC reduction. The level of monthly DOT of cephalexin increased from Phases 3 to 4. This finding was attributed to the educational lecture and pop-up displays. From Phases 2 to 3, the trend in the monthly DOT for amoxicillin and amoxicillin-clavulanate increased, which was considered an effect of ID consultation and ASP. The increase in the cephalexin, amoxicillin, and amoxicillin-clavulanate DOT indicated appropriate antimicrobial use. Furthermore, for trimethoprim-sulfamethoxazole, the trend increased from Phases 3 to 4, but the trend and level were not consistent during the study period. Patients with cancer are treated with substantial amounts of glucocorticoids as antiemetics during chemotherapy to

**Table 2. Average purchase costs of 3GCs and all oral antimicrobials per 100 patient-days from Phases 1–4.**

| Phase | [a]Actual cost of 3GCs (USD) | [b]Adjusted cost of 3GCs (USD) | [a]Actual cost of all oral antimicrobials (USD) | [b]Adjusted cost of oral antimicrobials (USD) |
|---|---|---|---|---|
| 1 | 2.00 | 2.30 | 22.08 | 44.46 |
| 2 | 0.98 | 1.19 | 16.96 | 34.64 |
| 3 | 0.42 | 0.61 | 20.53 | 37.62 |
| 4 | 0.10 | 0.16 | 23.31 | 39.18 |

[a]Actual cost includes the cost of switching to generic drugs and the factors related to changes in drug prices.
[b]Adjusted cost was calculated based on drug prices in April 2020.

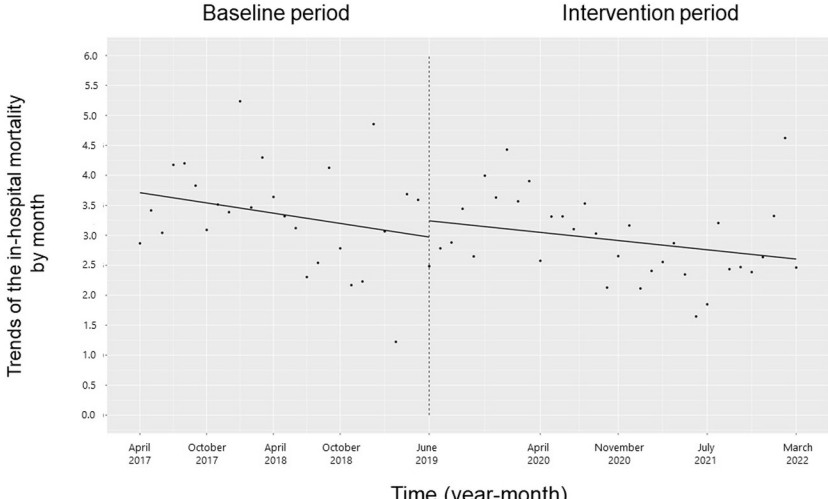

**Fig 12. Trends in in-hospital mortality by month during the pre-intervention period (Phase 1) and interventional period (Phases 2–4).** Each dot refers to the in-hospital mortality each month, and the slope is based on the linear regression in the four phases: Phase 1 (pre-intervention period from April 1, 2017, to May 31, 2019); Phase 2 (review of clinical pathways from June 1, 2019, to March 31, 2020); Phase 3 (establishing an infectious disease (ID) consultation service and implementing the antimicrobial stewardship program from April 1, 2020, to June 30, 2021); and Phase 4 (educational lecture and pop-up displays for oral antimicrobials from July 1, 2021, to March 31, 2022).

treat immune-related adverse events associated with immune checkpoint inhibitors and drug-induced pneumonia. This is a high-risk factor for *Pneumocystis jirovecii* pneumonia and a possible reason for the frequent use of trimethoprim-sulfamethoxazole for its prevention and treatment in our cancer center [29]. The monthly DOT for macrolides showed a decreasing trend from Phases 1 to 2 and 3 to 4, and the level decreased from Phases 2 to 3. The change

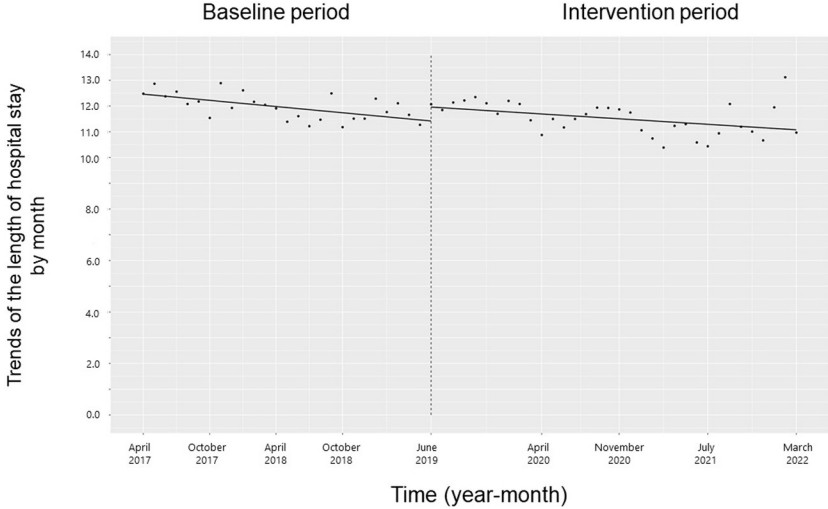

**Fig 13. Trends in the length of hospital stay by month during the pre-intervention period (Phase 1) and interventional period (Phases 2–4).** Each dot refers to the length of hospital stay each month, and the slope is based on linear regression in the four phases: Phase 1 (pre-intervention period from April 1, 2017, to May 31, 2019); Phase 2 (review of clinical pathways from June 1, 2019, to March 31, 2020); Phase 3 (establishing an infectious disease (ID) consultation service and implementing the antimicrobial stewardship program from April 1, 2020, to June 30, 2021); and Phase 4 (educational lecture and pop-up displays for oral antimicrobials from July 1, 2021, to March 31, 2022).

from Phases 2 to 3 was believed to be due to the impact of ID consultation and ASP, supporting the appropriate use of antimicrobials. Macrolides are commonly used for the treatment of atypical pneumonia, non-tuberculous mycobacterial infections, and sexually transmitted infections, but our hospital had few cases of these infections, and most of them were considered inappropriate for macrolide treatment. These findings for macrolide use during the study period were a favorable outcome, as they indicated that no switch was made from oral 3GCs. There was no change in the use of quinolone during the study period, implying that a simple switch from oral 3GCs did not occur. The use of quinolones could be attributed to their constant use in the treatment of low-risk febrile neutropenia [30] and to some physicians who do not follow the recommendations of ID physicians and antimicrobial stewardship teams.

The main concept of appropriate antimicrobial use in any patient population, including patients with cancer, is to ensure that each patient receives the most effective and safest antimicrobial agents for the treatment of infections, while simultaneously minimizing the impact on the ecosystem [31]. Patients with cancer have a high incidence of infections and require the use of antimicrobials for both treatment and prophylaxis, resulting in a significant amount of antimicrobial use [28]. This imposes significant antimicrobial pressure not only on the normal microflora of patients but also on the surrounding environment. The present study showed a significant reduction in the use of oral 3GCs without an increase in alternative broad-spectrum oral antimicrobials, but not the use of ESBL-producing Enterobacteriaceae, PRSP, BLNAR, CDI, or MRSA. A single-center study by Uda et al. showed that a reduction in the use of oral 3GCs did not reduce the incidence of PRSP and BLNAR, which is consistent with our current findings [21]. Similarly, Kato et al. showed that a reduction in the use of oral 3GCs did not change the incidence of ESBL-producing Enterobacteriaceae, MRSA, or AmpC beta-lactamase-producing bacteria [32]. There may be no association between the use of oral 3GCs and the incidence of these resistant organisms. However, the incidence of these resistant organisms in the pre-intervention period was most likely underestimated because the total number of hospitalized patient specimens in the current study increased significantly after the intervention. Our intervention did not result in significant changes in in-hospital mortality or length of stay, suggesting that it is safe and does not negatively impact patient outcomes. As these were single-center studies, future investigation is necessary to determine whether reducing the consumption of 3GCs in the community will reduce the prevalence of resistant microorganisms or improve patient outcomes.

Our intervention led to a reduction in the cost of 3GCs and the cost of adjusted purchases of all oral antimicrobials, which has economic benefits. The actual cost of purchasing all oral antimicrobials decreased significantly after the clinical path review but increased subsequently. This finding was most likely the result of increased appropriate use of alternative antimicrobial agents.

It is challenging to change the antimicrobial-prescribing behavior of physicians for several reasons: physicians tend to disregard the seriousness of preventing the development of resistant bacteria, and knowledge outside the physician's field of expertise does not appear to be updated [24]. Prospective audit and feedback of antimicrobials are effective but time-consuming [3], and intervention is often difficult because oral antimicrobials are already prescribed as a discharge prescription. The number of ID physicians in Japan is inadequate compared to that in the United States [3]. Furthermore, the training of ID physicians, pharmacists, laboratory technicians, and infection-control nurses in cancer centers is often time-consuming, given the complex background of patients with cancer, which requires sufficient experience and knowledge of ID and oncology. However, review of clinical pathways, educational lectures, and pop-up displays at the time of prescribing oral 3GCS may be relatively easy to address even in the absence of experts.

There are several limitations to this study. First, it is a single-center study in a Japanese cancer center; therefore, it is unclear whether a hospital-based oral 3GC reduction strategy can be generalized. As such, a long-term multicenter study including other Japanese cancer centers is warranted. However, the positive results at our institution, with its high antimicrobial-prescribing and large number of immunocompromised patients (patients with cancer) are likely to be adaptable to other hospitals. Second, the impact of antimicrobial prescriptions on patients with coronavirus disease (COVID-19) and hospitalized patients with suspected COVID-19 during the pandemic needs to be considered [33]. This is because antimicrobials have been used both prophylactically and therapeutically in such patients [33, 34]. That said, our hospital was affected by the COVID-19 pandemic, but the prescription of 3GCs showed a decline.

In conclusion, this is the first study to report on the impact of an oral 3GC reduction strategy in patients with cancer. We conducted a clinical path review, implemented an ID consultation and ASP, and provided educational lectures and pop-up displays at a cancer center in Japan. Our intervention reduced the use of oral 3GCs without worsening patient outcomes or increasing the use of alternative broad-spectrum antimicrobials and reduced the overall cost of oral antimicrobials. Overall, our strategy indicates that even facilities which use antimicrobials substantially can efficiently and easily reduce the use of 3GCs.

## Supporting information

**S1 Dataset.**
(XLSX)

## Acknowledgments

We are grateful to the clinical staff of the Aichi Cancer Center Hospital for their commitment toward providing patient care.

## Author Contributions

**Conceptualization:** Naoya Itoh.

**Data curation:** Naoya Itoh, Nana Akazawa, Daichi Kawamura, Hiromi Murakami, Yuichi Ishibana.

**Formal analysis:** Takanori Kawabata.

**Funding acquisition:** Naoya Itoh.

**Methodology:** Naoya Itoh.

**Supervision:** Eiichi N. Kodama, Norio Ohmagari.

**Writing – original draft:** Naoya Itoh, Takanori Kawabata, Nana Akazawa, Daichi Kawamura, Hiromi Murakami, Yuichi Ishibana, Eiichi N. Kodama, Norio Ohmagari.

**Writing – review & editing:** Naoya Itoh, Takanori Kawabata, Nana Akazawa, Daichi Kawamura, Hiromi Murakami, Yuichi Ishibana, Eiichi N. Kodama, Norio Ohmagari.

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
