## [Decision Letter · Decision Letter 0]

19 Dec 2022

PONE-D-22-26924Reduction strategies for inpatient oral third-generation cephalosporins at a cancer center: an interrupted time-series analysisPLOS ONE

Dear Dr. Itoh,

Thank you for submitting your manuscript to PLOS ONE. After careful consideration, we feel that it has merit but does not fully meet PLOS ONE’s publication criteria as it currently stands. Therefore, we invite you to submit a revised version of the manuscript that addresses the points raised during the review process.

The language needs improvement. 

We look forward to receiving your revised manuscript.

Kind regards,

Iddya Karunasagar

Academic Editor

PLOS ONE

Journal Requirements:

2. Please provide additional details regarding participant consent. In the ethics statement in the Methods and online submission information, please ensure that you have specified the need for consent was waived by the ethics committee, please include this information

"No"

Additional Editor Comments (if provided):

The language needs improvement

Reviewers' comments:

Reviewer's Responses to Questions

**Comments to the Author**

1. Is the manuscript technically sound, and do the data support the conclusions?

Reviewer #1: Yes

2. Has the statistical analysis been performed appropriately and rigorously? 

Reviewer #1: Yes

3. Have the authors made all data underlying the findings in their manuscript fully available?

Reviewer #1: Yes

4. Is the manuscript presented in an intelligible fashion and written in standard English?

Reviewer #1: Yes

5. Review Comments to the Author

Reviewer #1: Dear authors,

This is a good study about AMS outcomes especially in a setting where antimicrobial agents are often used out of fear to prevent and treat infections.

It would be useful to reduce the number of times the phases of the study are repeated in the result section.

6. PLOS authors have the option to publish the peer review history of their article (what does this mean?). If published, this will include your full peer review and any attached files.

Reviewer #1: **Yes: **Aruna Poojary

---

## [Author Response · Author response to Decision Letter 0]

4 Jan 2023

Our point-by-point responses to the comments and suggestions by Reviewer 1 are listed below.

We thank the reviewer for reviewing our manuscript and for the insightful comment that has helped us to significantly improve our manuscript. Although not an issue pointed out by the reviewer, we found an error in the description of the monthly DOT for macrolides in Phases 3 to 4 and revised the relevant passages, but it did not affect the main subject. Please note that our changes in the manuscript are highlighted in yellow.

Comment:

It would be useful to reduce the number of times the phases of the study are repeated in the result section.

Response:

We thank the reviewer for this comment. We removed statistically insignificant findings from the "results section" that duplicated those in the "figure legends" section as follows (Results, Page 10-20).

There were no significant changes in trends and levels from Phases 2 to 3 or from Phases 3 to 4. (Page 10, Lines: 229–230)

Each dot refers to the third-generation cephalosporins (3GCs) per 100 patients each month, and the slope is based on the linear regression in the four phases: Phase 1 (pre-intervention period from April 1, 2017, to May 31, 2019); Phase 2 (review of clinical pathways from June 1, 2019, to March 31, 2020); Phase 3 (establishing an infectious disease (ID) consultation service and implementing the antimicrobial stewardship program (ASP) from April 1, 2020, to June 30, 2021); Phase 4 (educational lecture and pop-up displays for oral antimicrobials from July 1, 2021, to March 31, 2022). (Fig 1, Page 11, Lines: 233–239)

There were no significant changes in trends and levels for cephalexin from Phases 1 to 2 or from Phases 2 to 3. The trend in the monthly DOT of cephalexin did not change from Phases 3 to 4, but the level increased (coefficients: 0.42; 95% CI: 0.03 to 0.81, p = 0.04; Fig. 2). (Page 11, Lines: 243–246)

There were no significant changes in the trend and level of the monthly DOT of amoxicillin and amoxicillin-clavulanate from Phases 1 to 2 (Fig. 3). The trend in the monthly DOT of amoxicillin and amoxicillin-clavulanate increased from Phases 2 to 3 (coefficients: 0.26; 95% CI: 0.07 to 0.45, p = 0.01) and decreased from Phases 3 to 4 (coefficients: -0.67; 95% CI: -0.89 to -0.45, p < 0.001), while the level did not change. (Page 11, Lines: 247–251)

The trend and level of the monthly DOT of sulfamethoxazole-trimethoprim did not change significantly from Phases 1 to 2 or Phases 2 to 3 (Fig. 4). However, the trend in the monthly DOT of sulfamethoxazole-trimethoprim increased from Phases 3 to 4 (coefficients: 0.26; coefficient: 0.38; 95% CI: 0.07 to 0.70, p = 0.02), while the level did not change. (Page 11, Lines: 252–255)

Each dot refers to cefalexin per 100 patients in each month, and the slope is based on the linear regression in the four phases: Phase 1 (pre-intervention period from April 1, 2017, to May 31, 2019); Phase 2 (review of clinical pathways from June 1, 2019, to March 31, 2020); Phase 3 (establishing an ID consultation service and implementing the ASP from April 1, 2020, to June 30, 2021); and Phase 4 (educational lecture and pop-up displays for oral antimicrobials from July 1, 2021, to March 31, 2022). (Fig 2, Page 12, Lines: 258–263)

Each dot refers to the amoxicillin and amoxicillin-clavulanate per 100 patients each month, and the slope is based on the linear regression in the four phases: Phase 1 (pre-intervention period from April 1, 2017, to May 31, 2019); Phase 2 (review of clinical pathways from June 1, 2019, to March 31, 2020); Phase 3 (establishing an ID consultation service and implementing the ASP from April 1, 2020, to June 30, 2021); and Phase 4 (educational lecture and pop-up displays for oral antimicrobials from July 1, 2021, to March 31, 2022). (Fig 3, Page 12, Lines: 266–272)

Each dot refers to the sulfamethoxazole-trimethoprim per 100 patients each month, and the slope is based on the linear regression in the four phases: Phase 1 (pre-intervention period from April 1, 2017, to May 31, 2019); Phase 2 (review of clinical pathways from June 1, 2019, to March 31, 2020); Phase 3 (establishing an ID consultation service and implementing the ASP from April 1, 2020, to June 30, 2021); and Phase 4 (educational lecture and pop-up displays for oral antimicrobials from July 1, 2021, to March 31, 2022). (Fig 4, Page 12, Lines: 275–281)

The trend and level of the monthly DOT of all three quinolones (moxifloxacin, ciprofloxacin, and levofloxacin) did not change significantly during the study period (Fig. 5). (Page 13, Lines: 284–285)

The trend in the monthly DOT of the three macrolides (erythromycin, azithromycin, and clarithromycin) decreased from Phases 1 to 2 (coefficients: -0.13; 95% CI: -0.22 to -0.04, p < 0.01), but the level did not change (Fig. 6). The trend in the monthly DOT of macrolides did not change from Phases 2 to 3 or from Phases 3 to 4, but the level decreased (coefficient: -0.96; 95% CI: -1.62 to -0.30, p < 0.01) and increased (coefficients: 0.92; 95% CI: 0.28 to 1.57, p < 0.01), respectively. (Page 13, Lines: 286–291)

Each dot refers to the quinolones per 100 patients in each month, and the slope is based on the linear regression in the four phases: Phase 1 (pre-intervention period from April 1, 2017, to May 31, 2019); Phase 2 (review of clinical pathways from June 1, 2019, to March 31, 2020); Phase 3 (establishing an ID consultation service and implementing the ASP from April 1, 2020, to June 30, 2021); and Phase 4 (educational lecture and pop-up displays for oral antimicrobials from July 1, 2021, to March 31, 2022). (Fig 5, Page 13, Lines: 294–299)

Each dot refers to the macrolides per 100 patients each month, and the slope is based on the linear regression in the four phases: Phase 1 (pre-intervention period from April 1, 2017, to May 31, 2019); Phase 2 (review of clinical pathways from June 1, 2019, to March 31, 2020); Phase 3 (establishing an ID consultation service and implementing the ASP from April 1, 2020, to June 30, 2021); and Phase 4 (educational lecture and pop-up displays for oral antimicrobials from July 1, 2021, to March 31, 2022). (Fig 6, Pages 13–14, Lines: 302–307)

The trend and level of the monthly incidence of ESBL-producing Enterobacteriaceae, PRSP, resistant H. influenzae and CDI did not change significantly during the study period (Figs. 7–10). The trend in the monthly incidence of methicillin-resistant Staphylococcus aureus (MRSA) increased during the study period (coefficients: 0.05; 95% CI: 0.04 to 0.07, p < 0.001), but the level did not change (Fig. 11). (Page 14, Lines: 311–315)

Each dot refers to ESBL-producing Enterobacteriaceae per 1000 patients in each month; the slope is based on linear regression in the four phases: Phase 1 (pre-intervention period from April 1, 2017, to May 31, 2019); Phase 2 (review of clinical pathways from June 1, 2019, to March 31, 2020); Phase 3 (establishing an infectious disease (ID) consultation service and implementing the antimicrobial stewardship program from April 1, 2020, to June 30, 2021); and Phase 4 (educational lecture and pop-up displays for oral antimicrobials from July 1, 2021, to March 31, 2022). (Fig 7, Page 14, Lines: 319–325)

Each dot refers to the PRSP per 1000 patients in each month; the slope is based on linear regression in the four phases: Phase 1 (pre-intervention period from April 1, 2017, to May 31, 2019); Phase 2 (review of clinical pathways from June 1, 2019, to March 31, 2020); Phase 3 (establishing an infectious disease (ID) consultation service and implementing the antimicrobial stewardship program from April 1, 2020, to June 30, 2021); and Phase 4 (educational lecture and pop-up displays for oral antimicrobials from July 1, 2021, to March 31, 2022). (Fig 8, Pages 14–15, Lines: 329–335)

Each dot refers to resistant H. influenzae per 1000 patients each month; the slope is based on linear regression in the four phases: Phase 1 (pre-intervention period from April 1, 2017, to May 31, 2019); Phase 2 (review of clinical pathways from June 1, 2019, to March 31, 2020); Phase 3 (establishing an infectious disease consultation service and implementing the antimicrobial stewardship program from April 1, 2020, to June 30, 2021); and Phase 4 (educational lecture and pop-up displays for oral antimicrobials from July 1, 2021, to March 31, 2022). (Fig 9, Page 15, Lines: 339–345)

Each dot refers to the CDI per 1000 patients each month, and the slope is based on linear regression in the four phases: Phase 1 (pre-intervention period from April 1, 2017, to May 31, 2019); Phase 2 (review of clinical pathways from June 1, 2019, to March 31, 2020); Phase 3 (establishing an infectious disease (ID) consultation service and implementing the antimicrobial stewardship program from April 1, 2020, to June 30, 2021); and Phase 4 (educational lecture and pop-up displays for oral antimicrobials from July 1, 2021, to March 31, 2022). (Fig 10, Pages 15–16, Lines: 349–355)

Each dot refers to MRSA per 1000 patients each month, and the slope is based on linear regression in the four phases: Phase 1 (pre-intervention period from April 1, 2017, to May 31, 2019); Phase 2 (review of clinical pathways from June 1, 2019, to March 31, 2020); Phase 3 (establishing an infectious disease (ID) consultation service and implementing the antimicrobial stewardship program from April 1, 2020, to June 30, 2021); and Phase 4 (educational lecture and pop-up displays for oral antimicrobials from July 1, 2021, to March 31, 2022). (Fig 11, Page 16, Lines: 359–365)

The actual (Phase 1 to 2: p < 0.001; Phase 2 to 3: p < 0.01, Phase 3 to 4: p < 0.001) and adjusted 3GC purchase costs (Phase 1 to 2: p < 0.001; Phase 2 to 3: p < 0.01, Phase 3 to 4: p < 0.01) per patient-days significantly decreased during the study period. (Page 16, Lines: 369–371)

There was no significant change in the trend of in-hospital mortality, length of hospital stay, or their level (Figs. 12 and 13). (Page 17, Lines: 392–393)

Each dot refers to the in-hospital mortality each month, and the slope is based on the linear regression in the four phases: Phase 1 (pre-intervention period from April 1, 2017, to May 31, 2019); Phase 2 (review of clinical pathways from June 1, 2019, to March 31, 2020); Phase 3 (establishing an infectious disease (ID) consultation service and implementing the antimicrobial stewardship program from April 1, 2020, to June 30, 2021); and Phase 4 (educational lecture and pop-up displays for oral antimicrobials from July 1, 2021, to March 31, 2022). (Fig 12, Page 18, Lines: 396–402)

Each dot refers to the length of hospital stay each month, and the slope is based on linear regression in the four phases: Phase 1 (pre-intervention period from April 1, 2017, to May 31, 2019); Phase 2 (review of clinical pathways from June 1, 2019, to March 31, 2020); Phase 3 (establishing an infectious disease (ID) consultation service and implementing the antimicrobial stewardship program from April 1, 2020, to June 30, 2021); and Phase 4 (educational lecture and pop-up displays for oral antimicrobials from July 1, 2021, to March 31, 2022). (Fig 13, Page 18, Lines: 405–411)

---

## [Decision Letter · Decision Letter 1]

25 Jan 2023

Reduction strategies for inpatient oral third-generation cephalosporins at a cancer center: an interrupted time-series analysis

PONE-D-22-26924R1

Dear Dr. Itoh,

We’re pleased to inform you that your manuscript has been judged scientifically suitable for publication and will be formally accepted for publication once it meets all outstanding technical requirements.

Kind regards,

Iddya Karunasagar

Academic Editor

PLOS ONE

Additional Editor Comments (optional):

All reviewer comments have been addressed satisfactorily

Reviewers' comments:

Reviewer's Responses to Questions

**Comments to the Author**

1. If the authors have adequately addressed your comments raised in a previous round of review and you feel that this manuscript is now acceptable for publication, you may indicate that here to bypass the “Comments to the Author” section, enter your conflict of interest statement in the “Confidential to Editor” section, and submit your "Accept" recommendation.

Reviewer #1: All comments have been addressed

2. Is the manuscript technically sound, and do the data support the conclusions?

Reviewer #1: Yes

3. Has the statistical analysis been performed appropriately and rigorously? 

Reviewer #1: Yes

4. Have the authors made all data underlying the findings in their manuscript fully available?

Reviewer #1: Yes

5. Is the manuscript presented in an intelligible fashion and written in standard English?

Reviewer #1: Yes

6. Review Comments to the Author

Reviewer #1: Dear Authors,

Thank you for addressing the previous comments to this manuscript. It addresses a very important topic on antimicrobial stewardship . Best wishes

7. PLOS authors have the option to publish the peer review history of their article (what does this mean?). If published, this will include your full peer review and any attached files.

Reviewer #1: **Yes: **Aruna Poojary

---

## [Editor Report · Acceptance letter]

31 Jan 2023

PONE-D-22-26924R1 

Reduction strategies for inpatient oral third-generation cephalosporins at a cancer center: an interrupted time-series analysis 

Dear Dr. Itoh:

I'm pleased to inform you that your manuscript has been deemed suitable for publication in PLOS ONE. Congratulations! Your manuscript is now with our production department. 

Kind regards, 

on behalf of

Dr. Iddya Karunasagar 

Academic Editor

PLOS ONE